# Cardiometabolic Risk Factors and Physical Activity Patterns Maximizing Fitness and Minimizing Fatness Variation in Malaysian Adolescents: A Novel Application of Reduced Rank Regression

**DOI:** 10.3390/ijerph16234662

**Published:** 2019-11-22

**Authors:** Zoi Toumpakari, Russell Jago, Laura D. Howe, Hazreen Abdul Majid, Angeliki Papadaki, Shooka Mohammadi, Muhammad Yazid Jalaludin, Maznah Dahlui, Mohd Nahar Azmi Mohamed, Tin Tin Su, Laura Johnson

**Affiliations:** 1Centre for Exercise, Nutrition, and Health Sciences, School for Policy Studies, University of Bristol, 8 Priory Road, Bristol BS8 1TZ, UK; russ.jago@bristol.ac.uk (R.J.); angeliki.papadaki@bristol.ac.uk (A.P.); laura.johnson@bristol.ac.uk (L.J.); 2MRC Integrative Epidemiology Unit, Population Health Sciences, Bristol Medical School, University of Bristol, Oakfield Grove, Bristol BS8 2BN, UK; laura.howe@bristol.ac.uk; 3Centre for Population Health, (CePH), Department of Paediatrics, Social and Preventive Medicine, Faculty of Medicine, University of Malaya 59100, 50603, Kuala Lumpur 50603, Malaysia; hazreen@ummc.edu.my (H.A.M.); shooka.mohammadi@gmail.com (S.M.); maznahd@ummc.edu.my (M.D.); nahar@ummc.edu.my (M.N.A.M.); tintin.su@monash.edu (T.T.S.); 4Department of Nutrition, Harvard Chan School of Public Health, Boston, MA 02115, USA; 5Department of Nutrition, Faculty of Public Health, University of Airlangga, Surabaya 60115, Indonesia; 6Department of Paediatrics, Faculty of Medicine, University Malaya, Kuala Lumpur 59100, Malaysia; yazidj@ummc.edu.my; 7South East Asia Community Observatory (SEACO), Jeffrey Cheah School of Medicine and Health Sciences, Monash University Malaysia, Bandar Sunway 47500, Malaysia

**Keywords:** adolescents, body mass index, cardiometabolic health, cardiorespiratory fitness, physical activity, physical activity patterns, trajectory, reduced rank regression

## Abstract

Patterns of physical activity (PA) that optimize both fitness and fatness may better predict cardiometabolic health. Reduced rank regression (RRR) was applied to identify combinations of the type (e.g., football vs. skipping), location and timing of activity, explaining variation in cardiorespiratory fitness (CRF) and Body Mass Index (BMI). Multivariable regressions estimated longitudinal associations of PA pattern scores with cardiometabolic health in n = 579 adolescents aged 13–17 years from the Malaysian Health and Adolescent Longitudinal Research Team study. PA pattern scores in boys were associated with higher fitness (r = 0.3) and lower fatness (r = −0.3); however, in girls, pattern scores were only associated with higher fitness (r = 0.4) (fatness, r = −0.1). Pattern scores changed by β = −0.01 (95% confidence interval (CI) −0.04, 0.03) and β = −0.08 (95% CI −0.1, −0.06) per year from 13 to 17 years in boys and girls respectively. Higher CRF and lower BMI were associated with better cardiometabolic health at 17 years, but PA pattern scores were not in either cross-sectional or longitudinal models. RRR identified sex-specific PA patterns associated with fitness and fatness but the total variation they explained was small. PA pattern scores changed little through adolescence, which may explain the limited evidence on health associations. Objective PA measurement may improve RRR for identifying optimal PA patterns for cardiometabolic health.

## 1. Introduction

Among adults, regular physical activity (PA) is associated with reduced risk of obesity, cardiovascular disease, type 2 diabetes and all-cause mortality [1]. In adolescents, PA is associated with improved cardiometabolic health, including lower levels of obesity and lower levels of insulin, glucose, lipids and blood pressure [2,3,4,5]. As PA levels track from adolescence into adulthood [6,7,8], establishing PA patterns that promote health in adolescence is important for future disease prevention.

The link between PA and cardiometabolic health is partially mediated by cardiovascular respiratory fitness (fitness) and obesity (fatness) [9,10,11]. Fitness is a function of genetic factors and the frequency of higher intensity PA improves cardiometabolic health [12]. In terms of obesity (indicated by Body Mass Index (BMI), the volume of PA is likely to be key for preventing excessive weight gain via increased total energy expenditure that is balanced with energy intake [13]. To maximize public health benefits, we need to identify the best combination of activities that optimize both fitness and BMI, as these concepts represent two independent pathways, via the intensity and volume of activity, from PA to cardiometabolic health.

The World Health Organization recommends that adolescents should engage in at least 60 min of moderate to vigorous intensity physical activity (MVPA) per day, with similar guidelines in many countries [14,15,16,17]. A limitation of these guidelines is the focus on the overall minutes of MVPA per day with limited guidance on the pattern of specific behaviors that adolescents should engage in, e.g., what activity, where or when. This represents a challenge for public health messages and hinders the translation from generic guidance to implementable recommendations. It is therefore imperative to identify specific patterns of activities that are optimally related to health outcomes, which could complement recommendations on overall minutes of MVPA per day. Characterizing patterns in different populations could provide context-relevant guidance on the ways in which adolescents should be active to improve their health. Such patterns would reflect types of activity (e.g., football vs. skipping), when (e.g., weekday vs. weekend) and where (e.g., school vs. after school) adolescents with a high fitness and low BMI are most commonly active—information that is crucial for guidance that relates to adolescents’ lived experiences. As these patterns of behavior are likely to be context-specific, i.e., adolescents in Malaysia engage in different patterns of behavior than adolescents in other countries, identifying such patterns in diverse countries is of vital importance.

Physical activity patterns can be derived using hypothesis-driven methods, e.g., indices that measure adherence to established physical activity guidelines, or entirely data-driven methods, e.g., to group participants with similar behavioral profiles together in terms of physical activity timing and sedentary behavior types [18]. Reduced Rank Regression (RRR) has been previously used in the derivation of dietary patterns associated with obesity and hypertension [19,20,21]. Combining data-driven and hypothesis-based approaches is potentially a more powerful technique to identify behavioral patterns that that are not only common but also related to disease [22]. RRR could therefore be used to derive activity patterns, which simultaneously maximize fitness and minimize fatness in adolescents. Such patterns could be used to create a priority list of types, timing and locations of PA for intervention development. These activities, times or locations could then form the basis of public health messages and interventions that are designed for specific contexts. To be effective, interventions should also focus on causal PA patterns that improve fitness, BMI and ultimately cardiometabolic health. Therefore, longitudinal studies are required to improve causal inference by identifying patterns of behavior that are associated with cardiometabolic health before risk develops. The aims of the current study were: (1) to apply RRR in order to identify potentially cardioprotective PA patterns for the first time with respect to the type, location and timing of PA, based on optimizing explained variation in fitness and BMI; (2) to explore pattern generalizability and tracking over 6 years during adolescence; and (3) to estimate longitudinal associations of a high fitness, low fatness PA pattern with cardiometabolic risk. We hypothesized that a high-fitness low-fatness pattern of PA would be associated with better cardiometabolic health during adolescence.

## 2. Materials and Methods

### 2.1. Sample and Study Design

Data are from 1828 adolescents repeatedly measured as part of the Malaysian Health and Adolescents Longitudinal Research Team study (MyHeARTs) at baseline (2012 (13 years)) and follow-up (2014 (15 years) and 2016 (17 years)). MyHeARTs has been described elsewhere [23]. Briefly, it is a prospective longitudinal observational cohort study aiming to prevent non-communicable diseases by identifying risk factors in adolescence. A two-stage stratified (8 urban and 7 rural schools) random sampling design of public schools in the central and northern regions of Peninsular Malaysia was used. All students in the first year of secondary school, who were able to read and write in Malay, were invited to participate. All participants, along with their parents, provided oral assent and written consent prior to participating. The study was approved by the Ethics Committee, University of Malaya Medical Centre (ethical approval number-14-376-20486).

### 2.2. Measurements

*Physical activity.* Physical activity type, timing and location was self-reported through a Malay version of the reliable and validated Physical Activity Questionnaire for older children (PAQ-C) [24,25], which has shown good internal consistency among Malaysian adolescents [26]. Ten items (Likert scales from 1 to 5) captured the frequency of various types of PA over the previous week (e.g., ‘Have you done any of the following activities in the past week?’); the frequency or intensity of PA at different times and locations including school, at home, during the week and weekend (e.g., ‘In the last 7 days what did you do most of the time at recess, lunch, after school, evenings, weekends?’).

#### 2.2.1. Cardiorespiratory Fitness (CRF) 

CRF was assessed with the Modified Harvard Step Test [27], an exercise test where participants step onto a 30 cm high step box, for a total of 5 min while their pulse is measured. Adolescents’ CRF score was calculated by multiplying the total duration of exercise (seconds) by 100, divided with the sum of three pulse readings (beats per minute) at 0–2 min of rest.

#### 2.2.2. Body Mass Index (BMI)

Height and weight were measured in light clothing. BMI (kg/m^2^) was calculated, standardized into z-scores for age and gender according to WHO growth reference data [28] and categorized to normal weight, overweight and obesity using international cut-off points from the International Obesity Task Force [29] for descriptive purposes.

#### 2.2.3. Cardiometabolic Risk Factors 

A fasting blood sample (15 mL) was collected at each time point from which plasma glucose (mmol/L), triglycerides (mmol/L) and LDL cholesterol (mmol/L) were measured. Systolic and diastolic blood pressure (BP) (mmHg) were measured as the mean of three readings from the right arm. Waist circumference (WC) (cm) was measured with an inelastic Seca measuring tape and percentage of body fat was measured with a portable body composition analyzer (Tanita, SC-240 MA).

#### 2.2.4. Potential Confounders 

Confounders were selected based on previously reported associations in the literature [4,5,30]. Urbanicity (urban vs. rural) was assessed based on the location of the school (criteria provided by the Department of Statistics in Malaysia). Adolescents self-reported on their gender, ethnicity, smoking status, having asthma and screen-time behaviors, i.e., TV and computer use (h/day). Pubertal development was self-assessed by Tanner stage pictures (stage 1–5) and sleep duration (average sleeping hours per day) were calculated from self-reported sleep and wake-up time. Diet was assessed through a 7-day diet history with the help of trained dietitians. A dietary pattern score explaining dietary energy density, fiber density and percentage of energy from fat was also derived using RRR at age 13 years (as done elsewhere) [19,31] and was used as a measure of overall diet. Parents self-reported their own education, employment and household income. The timing of all measures is presented in Appendix A.

### 2.3. Statistical Analysis

The dataset used and analyzed during the current study is available from the MyHeARTS team in the University of Malaya. All analyses (Appendix A) were conducted in Stata (15.1, StataCorp, Texas, USA) and RRR was run in SAS (9.4, SAS Institute, North Carolina, USA). Survey weights, provided by MyHeARTs, were used in all analyses to account for participants’ non-selection and non-response. The use of survey weights means that percentages (with 95% confidence intervals) are presented rather than absolute frequencies.

#### 2.3.1. PA Pattern Score

Reduced Rank Regression (RRR) [22] generated a PA pattern score. This was a weighted linear combination of responses to PAQ-C questions, by optimizing variation in specified intermediate variables hypothesized to be on the pathway PA and cardiometabolic health [11,32,33,34,35,36]. CRF [35] and BMI [32,34] were selected as intermediate variables based on previous associations reported in adolescents [11,33]. The RRR model included 23 predictor variables, i.e., 17 types of PA and six variables regarding timing and location of PA. The first generated PA pattern explained 6.4% variation in both fitness and BMI, whereas the second explained 1.8%. Therefore, only the first PA pattern was retained for subsequent analyses. A pattern score was calculated for each adolescent as the weighted sum of responses for each PAQ-C predictor multiplied by its pattern loading. A pattern loading can be interpreted as a correlation of each predictor variable with the underlying PA pattern. PA pattern scores at 15 years and 17 years were derived with confirmatory RRR, which used the PAQ-C responses at 15 years or 17 years multiplied by the pattern loadings generated using CRF and BMI data from 13 years only.

Pattern comparability across 13 years population subgroups defined by gender, ethnicity, urbanicity and age was assessed by performing a new RRR in each subgroup and comparing pattern loadings with those derived in the whole sample at 13 years using Tucker’s coefficient of congruence (φ) [37]. A φ < 0.85 indicates that the pattern loadings are different and hence the PA pattern is not generalizable from the entire population to that subgroup and therefore different PA patterns should be used for association analysis [37]. Owing to sex differences in the pattern loadings, sex-specific patterns were subsequently computed, and association analyses were conducted separately for boys and girls.

Longitudinal associations between change in the PA pattern score and cardiometabolic health at 17 years were examined with a two-stage approach, which was used to (1) summarize sex-specific trajectories of the PA pattern score from 13 to 17 years and (2) examine whether these PA trajectories are associated with cardiometabolic health at 17 years. In stage one, individual PA pattern score trajectories (n_boys_ = 675, n_girls_ = 1043) were derived from sex-specific, linear, three-level models (level 1: timepoints 13 years, 15 years and 17 years, level 2: adolescent, and level 3: school) with PA pattern score at age 13, 15 and 17 as the outcome, adjusted for age, ethnicity and urbanicity. Tracking of the PA pattern score was estimated with the intra-class correlation coefficient (ICC) in the final model (see Appendix A for more information).

In stage two, associations between baseline (13 years) and change in PA pattern score (13–17 years) with each cardiometabolic health outcome at 17 years separately were estimated using sex-specific linear regressions, adjusted for ethnicity, urbanicity, dietary pattern score and cardiometabolic health outcomes at 13 years (sample size: n_boys_~195, n_girls_~465 (Appendix A). Standardized beta coefficients are reported to enable direct comparisons of the association estimates across different health outcomes. Regressions were conducted on unrestricted and complete case samples (see Appendix A for more information).

Change over time in CRF score, BMI and health outcomes was examined with repeated measures ANOVA and mean change per year was calculated by averaging the individual mean changes between 13–15 years and 15–17 years. Pearson’s partial *r* correlations (adjusted for age) were calculated to assess the relationship between CRF, BMI and the PA pattern score. To explore trends in PA pattern score across different adolescent characteristics, sex-specific marginal means of the PA pattern score, adjusted for ethnicity and urbanicity (except for when ethnicity and urbanicity, which were adjusted for each other), were computed. Absolute BMI values were used in gender-specific analysis for adolescents of the same age, as systematic variation in BMI by age and sex is removed.

#### 2.3.2. Sensitivity Analyses

We also investigated whether intermediate variables (CRF and BMI) were individually associated with cardiometabolic health, cross-sectionally and longitudinally. For this, we used sex-specific, linear regressions (adjusted for ethnicity, urbanicity and dietary pattern score), with CRF or BMI as the independent variable and each cardiometabolic health outcome separately. For longitudinal associations, average change in CRF and BMI per year were used as the main exposure predicting cardiometabolic risk factors at 17 years, additionally adjusted for baseline risk factors at 13 years.

## 3. Results

Overall, 1333 adolescents participated at age 13 years, 1192 at 15 years and 1034 at 17 years. Information about the sample recruited and used at each stage of analysis is presented in Appendix A. Characteristics of adolescents at 13 years are presented in Table 1. Adolescents were mainly girls (64%), of Malay origin (79%) and attending urban schools (69%). In terms of their socio-economic profile, most adolescents had a high household income (62%), mothers with a high education (67%) who were homemakers (57%) and fathers with a high education (66%) who had a full-time jobs (81%). At 13 years, most adolescents had normal weight (53%), a marginally acceptable level of fitness (48%), were non-smokers (92%), did not have asthma (94%), were classified as Tanner Stage 4 (55%), slept more than 8.4 h/day (52%), watched TV more than 2.5 h/day (51%) and used a computer for fewer than 1.6 h/day (68%). Compared with the overall sample, the sample with complete data were more likely to be girls (73% vs. 64%), from rural areas (41% vs. 31%), with a low household income (48% vs. 39%) and underweight (15% vs. 22%).

Table 2 presents descriptive statistics for CRF score, BMI and cardiometabolic health outcomes in boys and girls. On average, there was an increase (per year) in boys’ CRF score (7.6 ± 8.0, *p* < 0.001), BMI (0.6 ± 0.9 kg/m^2^, *p* < 0.001) and WC (0.7 ± 7.6 cm, *p* < 0.001), whereas their serum glucose (−0.2 ± 0.6 mmol/L, *p* < 0.001), LDL cholesterol (−0.2 ± 0.6 mmol/L, *p* < 0.001) and percentage of body fat (−3.4 ± 10.6, *p* < 0.001) decreased. A similar pattern was observed among girls who also increased their CRF score, however to a lesser extent than boys (3.4 ± 8.3, *p* < 0.001), as well as their BMI (0.6 ± 0.7, *p* < 0.001). There was evidence for change across all girls’ cardiometabolic health outcomes; blood pressure (−0.8 ± 14 mmHg, *p* < 0.001), serum glucose (by −0.1 ± 0.4 mmol/L, *p* = 0.02) and triglycerides (−0.1 ± 0.4 mmol/L, *p* < 0.001) decreased, whereas waist circumference (3.7 ± 6.3cm, *p* < 0.001) and percentage of body fat (3.1 ± 5.7, *p* < 0.001) increased.

The cross-sectional PA pattern that we generated using RRR at 13 years explained 6.4% of the total variation in both fitness and BMI, however it explained more variation in fitness (12.3%) than BMI (0.6%) (Appendix A). The PA pattern score at 13 years was moderately correlated with fitness (r = 0.4, *p* < 0.001) and weakly correlated with BMI (r = 0.04, *p* = 0.106) (Appendix A). Population-subgroup PA patterns according to ethnicity, urbanicity and age were considered equivalent to the overall pattern of adolescents at 13 years (φ range = 0.88–0.99) (Appendix A). In contrast, PA patterns in boys and girls were different compared to the overall sample (φ 0.84 and 0.79 respectively). Specifically, the PA pattern in boys explained a similar amount of total intermediate variation (6.7%) but accounted for half the variation in fitness (6.6%), and ten times the variation in BMI (6.8%) compared to the overall sample. In girls, the PA pattern explained only half the variation in intermediates (3.3%), equating to half the variation in fitness (6.4% vs. 12.3%), but explained similar variation in BMI as in the overall sample (0.2%). The pattern was moderately correlated with higher fitness (r = 0.3, *p* < 0.001) and lower BMI (r = −0.3, *p* < 0.001) in boys, and thus a higher score represents a high-fitness and low-fatness pattern of PA. For girls, the pattern was moderately correlated with higher fitness (r = 0.4, *p* < 0.001) and weakly correlated with lower BMI (r = −0.1, *p* = 0.04), and thus the girls’ score primarily represents a high fitness pattern of PA.

The activities characteristic of boys’ and girls’ PA patterns are illustrated in order of the pattern loadings in Figure 1. Higher frequencies of jogging/running (pattern loading = 0.46), being very active in the evening (pattern loading = 0.37), at recess (pattern loading = 0.36), in the weekend (pattern loading = 0.31) and cross-country (pattern loading = 0.29) were associated with a higher pattern score in boys. In girls, bicycling (pattern loading = 0.54), tag (pattern loading = 0.38), being active in the evening (pattern loading = 0.33), after school (pattern loading = 0.32) and at PE class (pattern loading = 0.26) were most associated with a higher pattern score.

Trends in boys’ and girls’ PA pattern score according to various characteristics are presented in Figure 2. On average, boys had a higher pattern score compared to girls, however there was no evidence for the PA pattern score being different across the different characteristics at 13 years or over time.

Appendix A shows adolescents’ predicted PA pattern score at 13 years and its change over time. The mean predicted PA pattern score among Malaysian boys living in urban areas at 13 years was 0.01 (95% CI −0.12, 0.14), while there was no change in the PA pattern score per year in the unadjusted or fully adjusted models (adjusted β = −0.01, 95% CI −0.04, 0.03). In Malaysian girls living in urban areas, their mean predicted PA pattern score at 13 years was lower compared to boys (−0.08, 95% CI −0.15, 0.001) and declined by −0.08 (95% CI −0.1, −0.06) per year, independently of their ethnicity and urbanicity. The ICC for both boys and girls in the fully adjusted Model 4 was 35%, indicating weak to moderate tracking of the PA pattern score over time (Appendix A).

Figures present marginal means from sex-specific survey weighted linear three-level models with PA pattern score as the outcome, and the individual different characteristics as the main exposure (adjusted for age, ethnicity and urbanicity, except for when ethnicity and urbanicity were adjusted for each other). For the dietary pattern score, quintiles were generated where DP5 represents a more obesogenic diet. Form 3: secondary education, Form 4: post-secondary education. Abbreviations: DP, Dietary Pattern Score.

Regression findings from our stage-two analysis, showing associations between change in PA pattern score (per year) and cardiometabolic health outcomes at 17 years in boys or girls, are presented in Figure 3. In both the unrestricted or complete case sample, there was no evidence for associations between change in PA pattern score (per year) and cardiometabolic health outcomes at 17 years in boys or girls. Similarly, there was no evidence for cross-sectional associations between the PA pattern score and cardiometabolic health outcomes in boys or girls (Appendix A respectively). In contrast, we observed that an increased CRF score was associated with lower waist circumference, serum glucose, LDL, and triglyceride concentrations in boys and a lower waist circumference and serum glucose, but not LDL, and triglyceride concentrations in girls (Figure 4). Additionally, a higher BMI at 13 years in both boys and girls was consistently associated with worse cardiometabolic health profiles, except for serum glucose among boys (Figure 4). Detailed cross-sectional and longitudinal associations between the intermediate variables and cardiometabolic health are presented in Appendix A for boys and Appendix A for girl.

## 4. Discussion

This is the first study that has applied RRR to PA data and generated a novel pattern describing the types, location, and timing of PA that explain most variations in fitness and fatness in Malaysian adolescents. The different patterns in boys and girls suggested that distinct activities are associated with fitness in boys and girls and variation in fatness is explained by PA patterns among boys but not girls. PA pattern scores in both boys and girls remained largely static over time and there was no evidence for an association between change in PA pattern scores and cardiometabolic health at 17 years, despite direct associations of the specified RRR intermediates (CRF and BMI) with cardiometabolic health. A) unrestricted sample n_boys_ = 224, n_girls_ = 464 and B) complete case sample n_boys_ = 162, n_girls_ = 417. Estimates (standardized betas) come from sex-specific multivariate linear regressions with change in PA pattern score per year (PA trajectory) as the independent variable and individual cardiometabolic health outcomes at 17 years. All models are adjusted for ethnicity, urbanicity, the predicted PA pattern score at 13, dietary pattern score at 13 years and cardiometabolic health outcomes at 13 years. Coefficients are interpreted as a one-unit change in PA pattern score per year that is associated with beta standard deviation change on the cardiometabolic health outcome at 17 years. Abbreviations: BF, Body Fat; BMI, Body Mass Index; CRF, Cardiorespiratory Fitness; DBP, Diastolic Blood Pressure; GLU, Glucose; LDL, Low-Density Lipoprotein; SBP, Systolic Blood Pressure; SD, Standard Deviation; TAG, Triglycerides; WC, Waist Circumference.

Estimates (standardized betas) come from sex-specific multivariate linear regressions with a) average change in CRF over time or b) change in BMI over time, as the independent variable and individual cardiometabolic health outcomes at 17 years. All models are adjusted for ethnicity, urbanicity, dietary pattern score at 13 years and cardiometabolic health outcomes at 13 years. Average change of CRF and BMI per year is computed by calculating the average change per year in two separate time periods, i.e., 13–15 years ((15 years–13 years)/2) and 15 years–17 years ((17 years–15 years)/2), and then computing their average. Coefficients are interpreted as a) one-unit change in CRF score per year that is associated with beta standard deviation change in cardiometabolic health outcome at 17 years. Abbreviations: BF, Body Fat; BMI, Body Mass Index; CRF, Cardiorespiratory Fitness; DBP, Diastolic Blood Pressure; GLU, Glucose; LDL, Low-Density Lipoprotein; SBP, Systolic Blood Pressure; SD, Standard Deviation; TAG, Triglycerides; WC, Waist Circumference.

Due to the novelty of applying RRR to PA, there are no other studies with which to directly compare our findings. However, RRR has previously identified obesogenic dietary patterns in Australian [38,39], European [39] and UK adolescents [19], where the % variation explained in all intermediates, such as dietary energy density, fiber density and percentage of energy from fat, is typically higher (31–58%) compared to our findings (<10%). Similar low levels of variation (5.7–17.2%) have been reported in diet-based RRR patterns when biomarkers were used as intermediates rather than nutrient intakes [40,41,42,43]. This may suggest that CRF score and BMI may be too distal in the hypothesized pathway. It is possible that incorporating objective measures, such as accelerometry assessment, of the intensity and total volume of PA (more proximal intermediates) may reduce noise and result in a pattern that explains more variation relevant to the theoretical pathway.

The PA pattern score remained relatively stable from 13 to 17 years, especially in boys. This contrasts with previous findings in Western countries, which have shown a distinct decline in PA throughout adolescence [44,45,46]. For example, in a similar cohort of UK adolescents (complete case sample—599 boys and 742 girls) from the Avon Longitudinal Study of Parents and Children (ALSPAC) study, objectively measured light PA declined by approximately 12% per year from 12 to 16 years [46]. This decline in adolescents is also supported by self-reported PA data. A pooled analysis of 22 longitudinal studies (mostly conducted in the USA) in children and adolescents 9–19 years showed a 7.3% (95% CI −9.3%, −5.3%) annual decrease in overall PA [45]. Moving to secondary school is a key transition point where PA declines, especially in girls [47], mainly due to changes in PA opportunities outside of school [48]. It may therefore be that the time period covered in our study is relatively stable for Malaysian adolescents, who are already enrolled in secondary school with no major changes in the environment, which results in little change in PA.

We observed evidence for associations between CRF score and BMI with health outcomes, which is in accordance with previous studies in youth [9,10,11]. This suggests that the proposed theoretical pathway and the specified intermediates were appropriate, but the pattern did not capture enough variation in fitness and BMI to enable us to detect any associations between the PA pattern score or change in the PA pattern score and cardiometabolic health. This may be due to the lack of sensitivity to change and precision of the mean PA [49]. Another possible explanation is that fitness is also determined by genetic factors, which may account for 50% of the variability in fitness-exercise response, independent of demographics, body weight and baseline fitness [12,50]. Hence, associations between fitness and cardiometabolic health in the current sample may primarily be determined by a genetic component rather than PA patterns. A lack of longitudinal associations may also be attributed to the little change in PA pattern score over time, resulting in limited power to detect such associations.

Similar weak longitudinal associations between objectively measured PA and metabolic traits (cholesterol, blood pressure, triglycerides and glucose) have been observed in UK adolescents aged 12–15 years participating in ALSPAC (n = 1826) (longitudinal association sizes < −0.08 standard deviation units) [51]. This has been attributed to the relatively healthy UK adolescent sample, which has a similar metabolic profile to Malaysian adolescents in the current study (differences only in SBP (107 vs. 123 mmHg) and LDL cholesterol (2.8 vs. 1.1 mmol/L) in MyHeARTs and ALSPAC respectively). Additionally, a short duration of the follow-up (3 years), similarly to our study (4 years), may limit the potential for observing associations with change. PA patterns may form over a longer period, and thus further follow-up is needed to assess its long-term impact on health.

Despite no observed associations between PA pattern score and health outcomes, our pattern identified key differences in the activities that explain fitness and fatness in boys and girls, which are important when considering the design of interventions in Malaysia.

### Strengths and Limitations

A major strength of the study is the novel application of RRR to PA data. Analyses were carried out in a large sample of adolescents from a longitudinal cohort, which allowed the examination of change in PA patterns in relation to future health outcomes. PA pattern score trajectories were generated using the random effects models, accounting for within and between person variation in PA pattern score, which contribute to differences in individual response in exercise [50]. Our analyses also benefited from a direct measure of fitness and an objective measure of BMI. The use of survey weights enables a more robust analysis by accounting for participants’ non-selection and non-response. Limitations include the self-reported measure of PA, which has been reported to have low validity [52] and may therefore not have been able to accurately capture variation in fitness and BMI. However, using self-reported measures of PA allowed us to understand patterns in the type, timing and location of PA, which is not currently possible with the use of objective measures, which tend to only measure the volume and intensity of PA. There was a degree of missing data, mainly in fitness and health outcomes, which reduced the sample available for analysis from 740 boys and 1088 girls to ~223 boys and ~464 girls (Appendix A). However, complete case analysis did not reveal any differences to the main findings. We accounted for potential confounders in our analyses. However, as with all observational studies, residual confounding cannot be ruled out.

## 5. Conclusions

RRR identified specific combinations of types, location, and timing of physical activities associated with fitness and fatness. CRF and BMI were associated with cardiometabolic health in these Malaysian adolescents. However, no associations between the PA pattern score and cardiometabolic health outcomes were observed. This may be because the variation explained in CRF and BMI was small and PA patterns changed little through adolescence. Future studies should combine accelerometry assessment with Global Positioning System (GPS) to objectively capture PA and the context in which it occurs.

## Figures and Tables

**Figure 1 ijerph-16-04662-f001:**
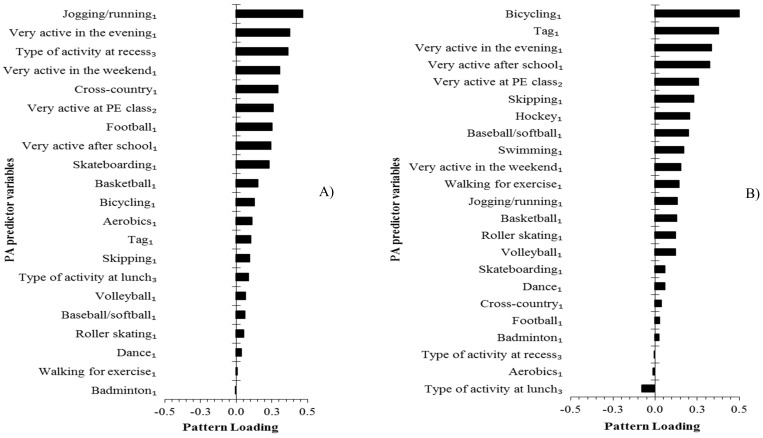
Physical activity (PA) pattern loadings for each predictor variable at age 13 years in (**A**) boys and (**B**) girls. _1_ Measured in times per week. _2_ Range 1 to 5—1: I don’t do PE, 2: Hardly ever, 3: Sometimes, 4: Quite often, and 5: Always. _3_ Range 0 to 5—1: Sat down, 2: Stood/walked around, 3: Ran/played a little bit, 4: Ran/played quite a bit, and 5: Ran/played hard most of the time.

**Figure 2 ijerph-16-04662-f002:**
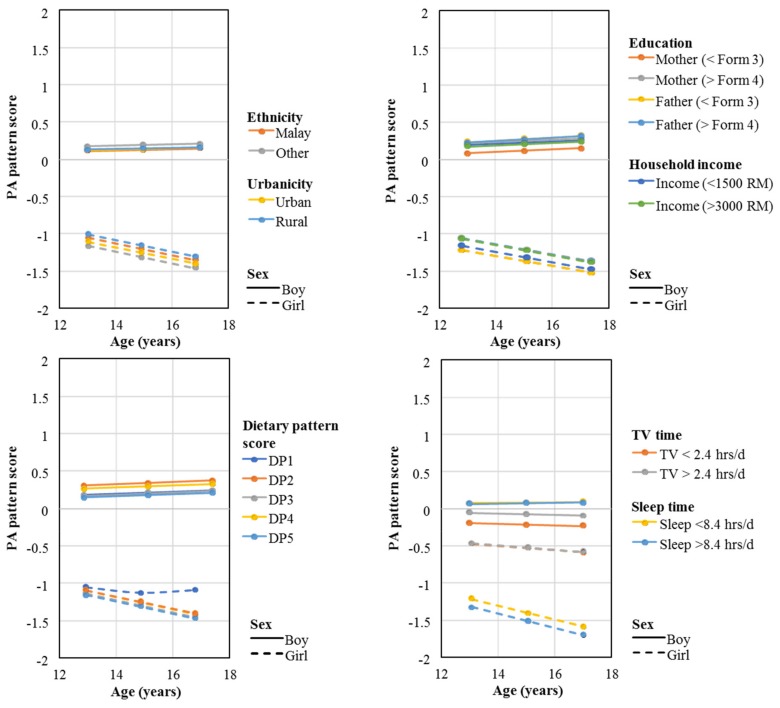
Average PA pattern score for different adolescent characteristics and by sex.

**Figure 3 ijerph-16-04662-f003:**
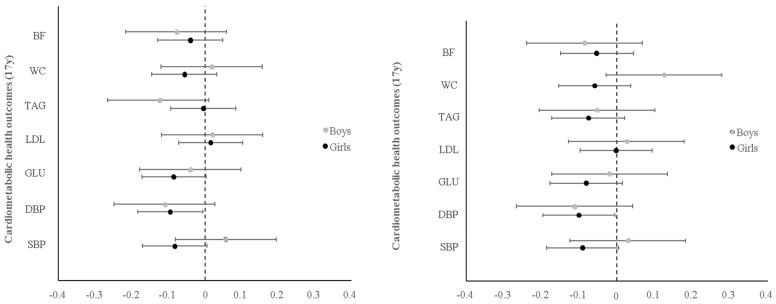
Associations between PA pattern score trajectories (13 years–17 years) and cardiometabolic health outcomes at 17 years in boys and girls.

**Figure 4 ijerph-16-04662-f004:**
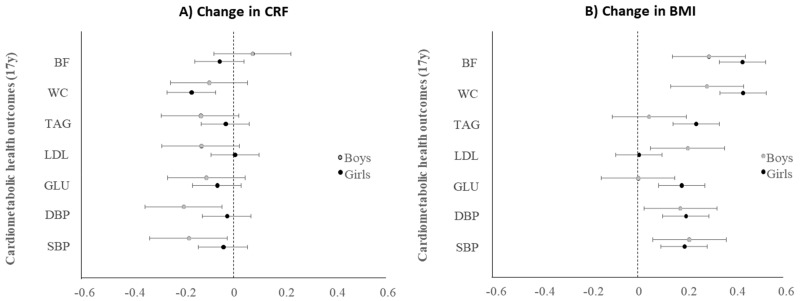
Associations between CRF and BMI trajectories (13 years–15 years) and cardiometabolic health outcomes at 17 years in boys and girls.

**Table 1 ijerph-16-04662-t001:** Description of adolescents at 13 years in the overall and smaller sample with all data required for longitudinal associations (complete case sample).

		Overall Sample N = 1333			Complete Case Sample N = 579		
		% N^1^	95% CI		% N^1^	95% CI	
**Socio-demographic variables**							
**Gender**	Male	36	30	43	27	20	35
	Female	64	57	70	73	65	80
**Ethnicity**	Malay	79	62	90	76	54	90
	Chinese	9	4	18	11	3	37
	Indian	10	3	25	9	2	27
	Other	3	1	9	4	1	12
**Place of school residence**	Urban	69	45	86	59	32	81
	Rural	31	14	55	41	19	68
**Mother’s education**	Form 3 and below	34	23	46	39	31	49
	Form 4 and above	67	54	77	61	51	69
**Father’s education**	Form 3 and below	34	23	48	39	28	52
	Form 4 and above	66	52	77	61	48	72
**Mother’s employment**	Homemaker	57	52	62	57	49	65
	Full-time work	35	30	41	33	26	40
	Part-time work	7	5	10	11	8	14
**Father’s employment**	Homemaker	8	6	11	8	6	12
	Full-time work	81	76	85	78	71	84
	Part-time work	11	9	14	13	9	19
**Household income**	<1500 RM	39	27	51	48	36	61
	1500–3000 RM	31	21	43	32	24	41
	>3000 RM	31	16	51	20	14	28
**Tanner Stage**	Stage 1–3	30	24	37	32	26	39
	Stage 4	55	50	60	54	49	60
	Stage 5	15	9	24	14	7	25
**Health variables**							
**Smokers**		8	5	12	6	5	7
**Having asthma**		6	4	9	6	3	9
**BMI**	Underweight	22	17	28	15	12	19
	Normal weight	53	51	56	54	51	58
	Overweight	17	14	20	23	20	27
	Obese	8	5	11	7	5	10
**CRF score**	Unacceptable (<65)	44	37	52	48	39	57
	Marginally acceptable (65–80)	48	40	54	44	34	53
	Acceptable (>80)	9	7	11	9	6	12
**Behavior variables**							
**Sleep**	≤8.4 h/day	48	41	54	52	40	63
	>8.4 h/day	52	46	59	48	37	60
**TV**	≤2.5 h/day	49	44	55	46	39	53
	>2.5 h/day	51	45	56	54	47	61
**Computer**	≤1.6 h/day	68	66	71	73	69	77
	>1.6 h/day	32	29	34	27	23	31

Abbreviations: BMI, Body Mass Index; CI, Confidence Interval; CRF, Cardiorespiratory fitness.^1^ Weighed percentages based on non-selection survey weights provided by MyHeARTs. There was 1% missing data in ethnicity, 13% in mother education, 14% in mother employment, 17% in father education, 20% in father employment, 12% in household income, 39% in Tanner stage, 1% in BMI, 17% in CRF score, 4% in sleep, 13% in TV and 23% in computer.

**Table 2 ijerph-16-04662-t002:** Descriptive statistics for Cardiorespiratory Fitness (CRF) score, Body Mass Index (BMI) and cardiometabolic health outcomes in boys and Girls.

Boys													
	13 years			15 years			17 years			Mean Change (Per Year)^1^			*P* ^2^
	N	Mean	SD	N	Mean	SD	N	Mean	SD	N	Mean	SD	
	433	71.4	12.9	422	72.0	12.1	346	103.3	19.0	325	7.6	8.0	<0.001
**BMI (kg/m^2^)**	507	20.0	5.5	442	20.7	5.3	348	21.3	5.2	361	0.6	0.9	<0.001
**SBP (mmHg)**	506	110.8	11.0	439	110.2	13.1	348	112.0	12.6	202	2.2	13.1	0.33
**DBP (mmHg)**	506	69.0	10.5	439	67.5	10.9	348	68.8	9.3	202	0.8	11.4	0.09
**GLU (mmol/L)**	509	5.0	0.6	452	4.8	0.5	347	4.8	0.6	202	−0.2	0.6	<0.001
**LDL (mmol/L)**	509	2.7	0.7	451	2.6	0.7	346	2.5	0.8	201	−0.2	0.6	<0.001
**TAG (mmol/L)**	509	0.9	0.5	452	0.9	0.5	347	0.9	0.7	202	0.1	0.7	0.05
**WC (cm)**	507	70.8	13.3	442	72.9	13.1	348	69.4	13.9	203	0.7	7.6	<0.001
**BF** (%)	506	19.0	14.7	442	15.7	11.2	348	14.3	9.2	202	−3.4	10.6	<0.001
**Girls**													
**CRF score**	674	61.1	12.5	713	55.9	16.5	679	73.2	19.1	620	3.4	8.3	<0.001
**BMI (kg/m^2^)**	808	19.9	4.6	724	21.6	5.1	685	22.0	5.2	648	0.6	0.7	<0.001
**SBP (mmHg)**	809	108.9	11.6	720	105.0	12.7	685	106.8	11.9	487	−0.8	14.0	<0.001
**DBP (mmHg)**	809	67.2	10.2	720	65.3	9.8	685	65.6	9.4	487	−0.5	12.1	0.03
**GLU (mmol/L)**	817	4.8	0.5	734	4.8	0.7	682	4.8	0.7	491	−0.1	0.4	0.02
**LDL (mmol/L)**	817	2.8	0.7	734	2.9	0.7	682	2.8	0.7	491	0.0	0.6	<0.001
**TAG (mmol/L)**	817	0.9	0.4	734	0.9	0.4	682	0.9	0.5	491	−0.1	0.4	<0.001
**WC (cm)**	808	67.9	10.4	724	71.6	11.2	685	71.3	10.8	487	3.7	6.3	<0.001
**BF** (%)	808	25.9	10.6	724	29.6	9.2	685	28.8	8.0	487	3.1	5.7	<0.001

Abbreviations: BF, Body Fat; BMI, Body Mass Index; DBP, Diastolic Blood Pressure; GLU, Glucose; LDL, Low-Density Lipoprotein; SBP, Systolic Blood Pressure; SD, Standard Deviation; TAG, Triglycerides; WC, Waist Circumference.^1^ Average change per year is computed by firstly calculating the average change per year in two separate time periods, i.e., 13–15 years ((15 years–13 years)/2) and 15 years–17 years ((17 years–15 years)/2) and then computing their average. ^2^ Repeated measures ANOVA.

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
