# Peer review of "Cardiometabolic Risk Factors and Physical Activity Patterns Maximizing Fitness and Minimizing Fatness Variation in Malaysian Adolescents: A Novel Application of Reduced Rank Regression"

_ijerph, 2019, doi:10.3390/ijerph16234662_

Round 1

Reviewer 1 Report

Overall, this is a clear, well-written paper that was a pleasure to read.

The introduction and discussion sections are very clear. The methods and results sections are lengthy/complex and difficult to follow.

Your methods section includes multiple layers of analyses - these need to be streamlined - is all of this content entirely needed (i.e., sections 2.3 and 2.4)? Perhaps restructure the section to improve clarity. As I read through ts multiple times in combination with the results it makes more sense but no reader wants to have to do this and it should simply be easier to follow in the first place. I think it may be the order with which you are presenting your analyses that causes the problem. It may also be that you are trying to squeeze too much into one paper.

The methods section lacks important details at times (see below).

Comments:

Abstract: Line 21 'patterns in the type, location and timing of PA'. 'Type' may mislead some readers - I assumed you were refering to walking or moderate or vigorous (i.e., intensity) not actually type of activity. I recommend you provide an example (as you do later in the text) to avoid reader misunderstanding - if you can find the space in terms of word count.

Introduction: Line 46-51, you speak of PA in relation to BMI, in particular - volume of PA and increased energy expenditure. Whilst this is true, energy intake (i.e., diet) is obviously also a key component of weight gain. This should be acknowledged here.

Line 56-57 'This represents a challenge to public health messages and hinders the translation from generic guidance to implementable recommendations' - this is an important point and justifies your study. I feel this gets lost a little in the paragraph so the importance of your study doesn't come across as well as it could (I mostly missed it on my first quick read through). If possible, add a little more here to bring out this point and the importance of your study.

Intro - last paragraph - Reduced Rank Regression - your reader is unlikely to be familiar with it (I wasn't). You provide references but a little more explanantion on the technique/approach may enable the reader to not have to go elsewhere to get a general understanding of the approach (e.g., similarities/differences to other methods - e.g., your ref 18 mentions PCA; and why you use RRR and not some other method). Give the reader a bit of a hand on this.

Methods - Physical activity - please provide a little more detail on the PAQ-C (e.g., reliability, validity, example of questions/question format) - as with the previous comment, your reader (and reveiwer) doesn't want to have to go and chase down references all the time if they are not already familiar with e.g., the questionnaire (or for RRR, technique).

Cardiometabolic risk factors - WC - elastic band? Please provide more detail as this does not sound reasonable (more commonly inelastic tape measure). %BF - please provide details of analyzer (e.g., brand, model number etc)

Potential confounders - provide detail on how dietary information was captured

Pattern score/statistics - These sections are  confusing as section 2.3 PA pattern score and CM health seems like it should belong within 2.4 Statistical analyses...These sections need some work to make them clearer to follow. Maybe include how the different pieces here relate to your aims? 

How was 'age' and/or 'time' treated in models? was this just 3 set timepoints (which is what it seems) or did you have continuous time? Were all participants measured with equal differences in time?

Line 154 'were associated' sounds like we are in the results - 'associations were estimated' (or similar) would be more appropriate

Your sample size for stage 2 (line 157) is substantially smaller than your initial sample - why is this? What are the predictors of missingness?  

line 170 - change over time in CRF score and BMI - should this be BMI z scores? It is not appropiate to use BMI longitudinally amongst children/adolescents. Given that your analyses include both cross-sectional and longitudinal BMI-Z score would be best to use throughout. Why have you not done this? Or have I misunderstood?

Results: line 198 - I am concerned at the use of BMI when reported change over time - given developmental growth this is not a good measure - why are you not using BMI z-scores?

Line 213 - I assume this is cross-sectionally? Please ensure this is clear throughout. It's implicit in your statement but it would be helpful if you made such things explicit given the complexity of your paper. Also - I assume this is reporting the results from the RRR - perhaps introduce the paragraph that way. You have performed a lot of pieces of analyses and some sign-posts for your reader may assist in following.

Please be consistent with presenting p values - you present them in text for Resutls paragraph 2 (lines 197-204), then in the first half of the paragraph starting line 213, but then you do not report them later in this paragraph. It is unclear which associations/correlations are therefore statistically significant and which are not.

Figure 1 - please clearly label to clarify which pertains to boys or girls

Line 258 - please begin this paragraph by relating back to your various analyses/aims to aid the reader. Why do you only look at change in PA in relation to CM outcomes? Why do you not also look at e.g., 13y PA to estimate change in (e.g., 13-15y) CM health? Or cross-sectional regression results (i.e., adjusted estimates of relationships) between PA and CRF/BMI (rather than just the correlations). Given that there is little change in PA scores over time - its unlikely that change would drive an effect large enough to be noticeable in your sample. There may be a cross-sectional relationship that remains relatively static over time though.

Discussion - line 300 'objectives'  should be 'objective'

line 303 - PA pattern score was relatively stable. How reflective of volume of PA was the pattern score? i.e., (and i may have misunderstood) the PA pattern has been constructed to maximise the variation in CRF and BMI at each time point - this differs from absolute PA volume. Did absolute volume of PA decline in your sample?  CRF increased over time overall - however this is likely influenced by development related to adolescence as well as PA participation - I note your trajectories across the time points are not obviously linear, its difficult with only 3 points). 

Line 343 - 'between person variation in PA' should more correctly be 'PA pattern score'

Line 352 were their difference between those with complete data v those with missing? particualrly those with missing fitness/health data?

Author Response

We tank the reviewer for their comments for our paper entitled ‘Cardiometabolic risk factors and physical activity patterns maximizing fitness and minimizing fatness variation in Malaysian adolescents: a novel application of reduced rank regression’. We were pleased to see that reviewers thought our manuscript is a clear and well-written paper with an interesting topic (Overall, this is a clear, well-written paper that was a pleasure to read).We were grateful for the opportunity to respond to the valuable comments and suggestions and have enclosed a point by point response detailing the changes we have made to our manuscript.

We hope that you will find that our responses address the main comments and we look forward to receiving your editorial decision.

Thank you for your consideration.

Yours sincerely

Zoi Toumpakari (on behalf of the authors)

Reviewer 2 Report

Personally I am not able to figure out the aims of the current study, more over the design is unclear (i.e. cross sectional, longitudinal, and if it is prospective or retrospective). I think it should be re-written properly, since the topic seems to be of interest. However I advise authors to take into account some of my below indications, for most to focus on a single aim rather than 3 or 4, and to present their data in a clearer and fluid manner.

The title should be shortened. The abstract include several abbreviation that make it difficult to read please improve the its presentation. I suggest that it should be re-written to include in one paragraph, a background on the topic, aim, methods, results and conclusion in a clear manner. The keyword should be keywords not statements, please adjust. The introduction is broad, should be more focused, and the aim too. Why not focus on one aim??? Moreover a hypothesis should be mentioned at the end of this section. The concept of RRR should be described and clarified in a simple way. The clinical implication of this study should be mentioned in the discussion section. The new directions of future research should be mentioned too. The presentation of table 1 is unclear. The English should be extensively edited, in general the statements should be shortened and go direct to the point. All in all the manuscript is too long including many speculations, authors should cut and reduce of its length. Moreover it is really dispersive in several of its section to become difficult to understand.

Round 2

Reviewer 2 Report

Authors were highly responsive. I think that the paper is suitable for publication in the current form.